# Periodontal pathogens alter the synovial proteome. Periodontal pathogens do not exacerbate macroscopic arthritis but alter the synovial proteome in mice

**Anna-Lena Buschhart**[1☯]**, Lennart Bolten**[1☯]**, Johann Volzke**[1]**, Katharina Ekat**[2]**, Susanne Kneitz**[3]**, Stefan Mikkat**[4]**, Bernd Kreikemeyer**[2]**, Brigitte Müller-Hilke**[1]*

1 Core Facility for Cell Sorting & Cell Analysis, Laboratory for Clinical Immunology, University Medical Center Rostock, Rostock, Germany, 2 Institute of Medical Microbiology, Virology and Hygiene, University Medical Center Rostock, Rostock, Germany, 3 Physiological Chemistry, Theodor Boveri Institute (Biocenter), University of Wuerzburg, Wuerzburg, Germany, 4 Core Facility for Proteome Analysis, Center for Medical Research, University Medical Center Rostock, Rostock, Germany

☯ These authors contributed equally to this work.
* brigitte.mueller-hilke@med.uni-rostock.de

**Data Availability Statement:** All relevant data are publicly accessible at ProteomeXchange (accession number: PXD020397).

## Abstract

Rheumatoid arthritis (RA) and periodontitis (PD) are chronic inflammatory diseases that appear to occur in tandem. However, the mutual impact PD exerts on RA and vice versa has not yet been defined. To address this issue, we set up an animal model and analyzed how two prime inducers of periodontitis—*Porphyromonas gingivalis (Pg)* and *Aggregatibacter actinomycetemcomitans (Aa)*–differ in their pathogenic potential. Our experimental setup included collagen induced arthritis (CIA) in the mouse, oral inoculation with *Pg* or *Aa* to induce alveolar bone loss and the combination of both diseases in inverted orders of events. Neither pathobiont impacted on macroscopic arthritis and arthritis did not exacerbate alveolar bone loss. However, there were subtle differences between *Pg* and *Aa* with the former inducing more alveolar bone loss if PD was induced before CIA. On a molecular level, *Pg* and *Aa* led to differential expression patterns in the synovial membranes that were reminiscent of cellular and humoral immune responses, respectively. The *Pg* and *Aa* specific signatures in the synovial proteomes suggest a role for oral pathogens in shaping disease subtypes and setting the stage for subsequent therapy response.

## Introduction

Rheumatoid arthritis (RA) is a systemic joint disease and a serious long-term illness that affects about 1% of the global population [1]. It is characterized by synovitis and deterioration of bone and cartilage, resulting ultimately in the loss of function of the affected joints [2, 3]. Treatment options to prevent irreversible damage include conventional synthetic and biological disease modifying anti-rheumatic drugs (DMARDs). As of yet, a range of biological DMARDs target different pathways of the inflammatory process driving the pathogenesis of

**Funding:** The author(s) received no specific funding for this work.

**Competing interests:** The authors have declared that no competing interests exist.

RA. However, independent of whether the TNFα-, T cell costimulatory-, B cell- or JAK/STAT signaling pathway is addressed, incomplete response rates hint at individually different pathogenic processes. As predictors of response to a given biological DMARD have not yet been defined, the majority of patients experience a phase of trial and error before disease activity is reduced or remission is achieved [4, 5].

RA is frequently associated with periodontitis (PD), a bacteria-driven chronic inflammatory ailment in humans affecting nearly 11% of the adults worldwide [6]. PD is characterized by dysbiosis of the oral microbiome which ultimately leads to chronic inflammation, progressive destruction of the periodontium and alveolar bone loss [7, 8]. Two oral pathobionts associated PD are *Porphyromonas gingivalis (Pg)* and *Aggregatibacter actinomycetemcomitans (Aa)* [9, 10]. While the former has been associated with the pathogenesis of chronic periodontitis in humans, the latter may be related to the severity of the disease [11]. Both came to the fore for their involvement in post translational modification of proteins which has been implicated in the breach of tolerance [12–15].

Both, the high incidence of PD among RA patients and severe PD accumulating in RA patients has spurred investigations not only into mutual responses to therapeutic interventions but also into common disease pathways [12, 16]. However, whether RA and PD share the same predispositions for severity and chronicity or whether they exacerbate each other is difficult to dissect in humans. We therefore turned to a mouse model to investigate the mutual impact of both diseases on each other. In particular, we were interested whether i) an existing joint disease exacerbates subsequent periodontal disease, ii) whether pre-existing periodontal disease promotes incidence and severity of subsequent arthritis and iii) whether *Pg* and *Aa* differ in their pathogenic potential for either arthritis or periodontal disease.

## Results

### Periodontal disease led to systemic immune responses and alveolar bone loss but was unaffected by subsequent arthritis

In order to investigate the impact periodontal disease (PD) exerts on subsequent collagen induced arthritis (CIA) or whether CIA exacerbates pre-existing PD, we designed an experimental setup whereby PD is induced prior to CIA (Fig 1). In short, an initial ten days of systemic antibiotics therapy was followed by a three days wash-out. Thereafter, prime inducers of citrullination—*Pg* or *Aa*–were repeatedly administered followed by a two weeks recovery phase. Application of sodium carboxy-methylcellulose in PBS served as sham treatment for PD induction. On experimental days 42 and 64, mice were primed and boosted using collagen type II in order to induce arthritis. After the boost, mice were regularly scored for incidence and severity of arthritis. Inflammatory cytokines were measured on experimental days 74 and at endpoint, while the assessment of joint histology was only done after sacrifice.

As the minute anatomy of the murine oral cavity did not allow for the assessment of acute gingivitis and periodontitis in live animals, we concentrated on antibodies against oral pathobionts and on systemic cytokines as proxies for successful colonization. In order to establish whether the gastrointestinal microbiome would serve as another proxy for colonization with oral pathobionts, we also monitored stool samples over the course of the experiments. Gingival histology and alveolar bone loss were assessed at endpoint.

Quantification of systemic antibodies against *Pg* and *Aa* revealed negative results for the sham yet clearly positive signals for the respective experimental groups (Fig 2A). Likewise, measuring inflammatory cytokines in the serum at experimental day 74 showed increased levels of IL-23, IL-1β, IL-27, IL-17A, INF-β but also IL-10 in mice that were inoculated with *Pg* and *Aa*, indicating a systemic response to the pathobionts (Fig 2B). Note, that CIA mice were

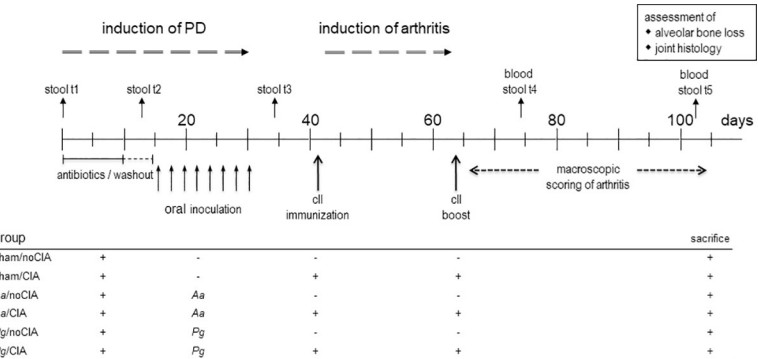

**Fig 1. The experimental setup of periodontal disease induced before CIA.** Periods for antibiotics treatment and macroscopic scoring of arthritis are indicated by horizontal lines and arrows, while individual days for oral application of the pathobionts, immunization, boost as well as blood and stool sampling are shown by vertical arrows. Experimental endpoint was day 105 and assessments of alveolar bone loss and joint histology were only made thereafter. Tables below the graphs summarize the individual experimental groups. *Pg*: *Porphyromonas gingivalis*; *Aa*: *Aggregatibacter actinomycetemcomitans*; sham: oral inoculation with PBS/ sodium carboxymethylcellulose.

not included in the cytokine analysis in order to prevent a blurring of results due to priming and boosting in the presence of Freund´s adjuvant. Analyzing the gastrointestinal microbiome revealed negligible changes around the time of oral application of pathobionts however, as

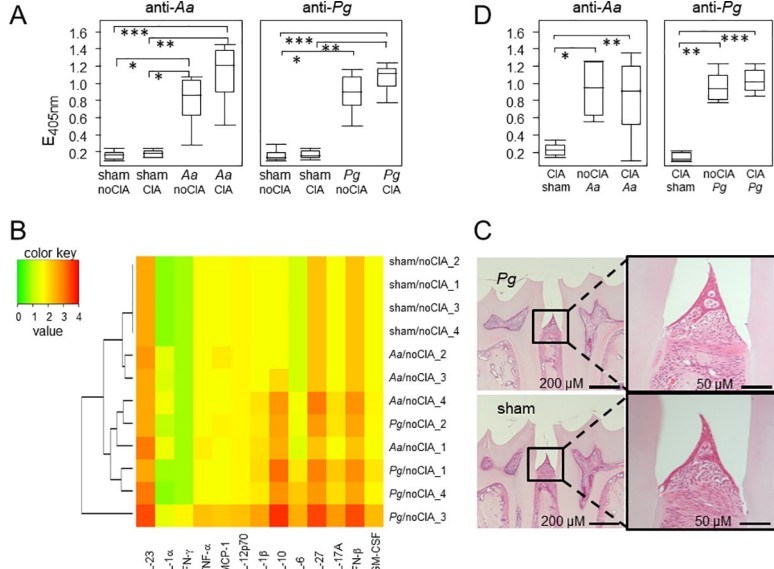

**Fig 2. Oral inoculation with pathobionts led to systemic immune responses.** (A) shows increased titers of antibodies against *Aa* (left) and *Pg* (right) in mice that were either sham treated (n = 7 for all sham groups) or orally inoculated with pathobionts (n = 8 for both *Aa* groups, n = 8 for *Pg*/noCIA and n = 9 for *Pg*/CIA). Statistical significance was calculated via Kruskal-Wallis test with post-tests and resulted in a p-value < 0.0001. Differences between the individual groups are indicated by asterisks. *p-value < 0.05, **p-value <0.01, ***p-value < 0.001. (B) The heat map summarizes the systemic expression levels of inflammatory cytokines (listed below the heat map) on day 74. The color key is shown on the left, individual animals are listed on the right and the dendrogram visualizes the connectedness between animals. Only n = 4 per group were analyzed. (C) shows examples of HE stained thin sections of jaws from mice that were either inoculated with *Pg* (upper panels) or sham treated (lower panels). (D) shows increased titers of antibodies against *Aa* (left) and *Pg* (right) in mice that were induced for CIA before PD (n = 8 for all groups). Statistical significance was calculated via Kruskal-Wallis test with post-tests and resulted in p-values = 0.0023 for anti-*Aa* and p = 0.0004 for anti-*Pg*. Differences between the individual groups are indicated by asterisks. *p-value < 0.05, **p-value <0.01, ***p-value < 0.001. Note that the sham/noCIA group is depicted in (A).

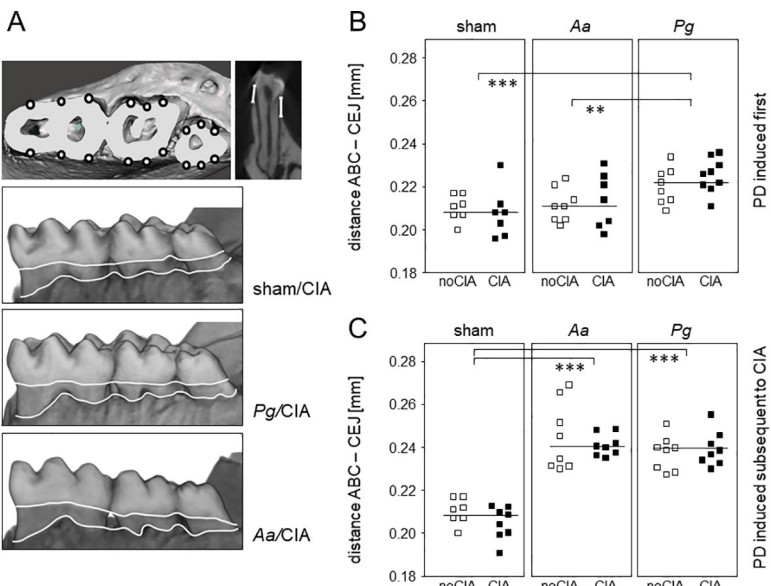

**Fig 3. Alveolar bone loss was unaffected by arthritis.** (A) 3D-models of the right hemi-mandible were calculated on the basis of µCT measurements and are here used to illustrate the sites for evaluating alveolar bone loss. Three geometrically distributed sites for the lingual and buccal sides of each tooth were defined, resulting in a total of 18 sites per hemi-mandible. At these 18 sites, the distances from the cemento-enamel junction (CEJ) to the alveolar bone crest (ABC) were measured in millimeters and the arithmetic means calculated. (B) Dot plots summarize alveolar bone loss as an increase in the ABC-CEJ distance for the various experimental groups in which PD was induced prior to CIA. Each dot represents one animal. One-way analysis of variance (ANOVA) resulted in a p-value of 0.0002, asterisks indicate in between group differences. *p-value < 0.05, ***p-value < 0.001. horizontal lines indicate means (C) Dot plots summarize alveolar bone loss as an increase in the ABC-CEJ distance for the various experimental groups in which PD was induced subsequent to CIA. Each dot represents the hemi-mandible of one animal. One-way analysis of variance (ANOVA) resulted in a p-value < 0.0001, asterisks indicate in between group differences. ***p-value < 0.001. Note that the animals receiving sham treatment in the absence of CIA are identical in (B) and (C).

both, *Porphyromonadaceae* and *Gammaproteobacteria* turned out to be highly abundant under physiological conditions, none of the changes observed did reach statistical significance (S1 and S2 Figs in S1 File). Likewise, histology of the mandibles at endpoint revealed no indication of inflammation (Fig 2C). In summary, successful inoculation with oral pathobionts was confirmed via antibodies and sustained increases in inflammatory cytokines, while acute inflammation of the gingiva or periodontium could not be sustained at endpoint.

Alveolar bone loss was quantified via imaging of the mandibles using micro-computed tomography (µCT) and calculating the distance between the cemento enamel junction (CEJ) and alveolar bone crest (ABC). Exemplary reconstructions are shown in Fig 3A. Fig 3B shows that inoculation with Pg resulted in significant alveolar bone loss which was absent in mice that were inoculated with *Aa*. Moreover, CIA by itself did not induce alveolar bone loss nor did it exacerbate PD related bone loss, as the distances between ABC and CEJ were comparable between CIA and noCIA groups. In summary, oral application of both pathobionts led to sustained immune responses including antibodies and inflammatory cytokines. In contrast, alveolar bone loss at endpoint was only detectable in mice inoculated with *Pg* and was unaffected by arthritis.

## PD did not impact on subsequent arthritis

Fig 4A and 4B show macroscopic and histologic pictures of arthritic and control paws. Inflammatory infiltrates which are characteristic of CIA are exemplified by histology. Edema (swelling) and redness as shown in the macroscopic image served to evaluate arthritis incidence and

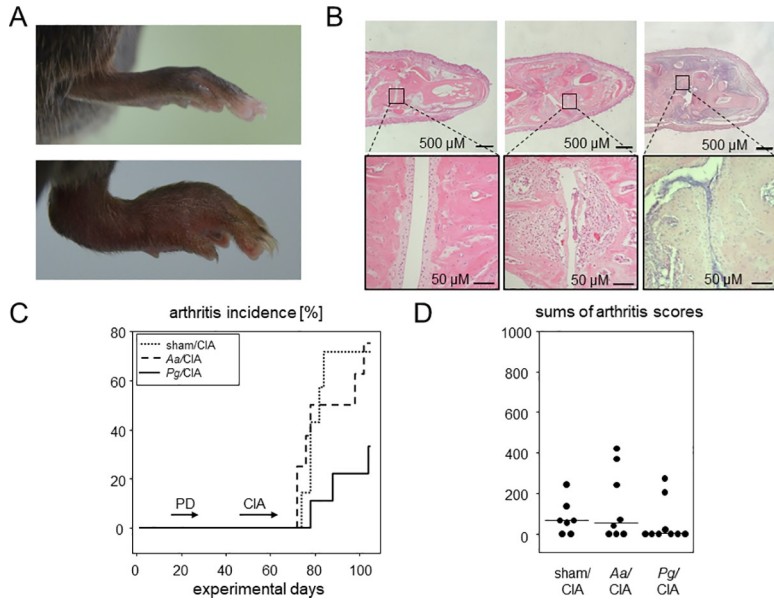

**Fig 4. PD did not impact on subsequent CIA.** CIA led to macroscopically visible swelling of the paws and microscopic deterioration of the joints (A) A swollen hind paw resulting from CIA (lower panel) in comparison to the paw of a control (sham/noCIA) mouse (upper panel). (B) HE stained thin sections of hind paws reveal inflammatory infiltrates in *Aa*/CIA and *Pg*/CIA compared to a control specimen from sham/noCIA animals. (C) Inverted Kaplan-Meier curves show arthritis incidences over time. Horizontal arrows indicate periods of disease inductions. Numbers are n = 7 for sham/CIA, n = 8 for *Aa*/CIA and n = 9 for *Pg*/CIA. (D) Dot plots summarize the sums of macroscopic arthritis scores at endpoint, assessed every other day starting from boost. Every dot represents a single mouse. CIA: collagen induced arthritis; PD: periodontal disease.

scores. When assessing the impact of pre-existing PD on CIA, there were no significant effects —neither for incidence, nor for severity (Fig 4C and 4D). Even though macroscopic disease development tended to develop later in mice that were orally inoculated with *Pg* compared to mice receiving *Aa* or sham treatment, the p-value resulting from a Breslow test was 0.086, indicating differences in the early phase of arthritis induction which however, did not quite reach significance. Arthritis incidences at endpoint ranged between 33.3 (*Pg*/CIA) and 75% (*Aa*/CIA). Cumulative arthritis scores were comparable between the sham, *Pg* or *Aa* groups, ruling out that PD exacerbated subsequent CIA.

## CIA slowed down the physiological weight gain and was associated with increased systemic cytokines

The physiological development as assessed via weight gain came to a halt during the period of PD induction when oral inoculations demanded repeated anaesthesia. Weight gain was resumed thereafter however, animals induced for arthritis gained weight at a slower rate than the non-CIA animals. At endpoint, the difference in weight was significant between the CIA and noCIA animals (Fig 5A). These significantly reduced weight gains were flanked by increased inflammatory cytokines. Hierarchical clustering based on serum levels of inflammatory cytokines at endpoint differentiated between mostly arthritic (CIA) and mostly non-arthritic (noCIA) mice (Fig 5B). Indeed, all cytokines except for IL-23 were most abundantly expressed in CIA mice. However, hierarchical clustering did not reveal any differentiation between sham treatment and inoculation with *Pg* or *Aa*, respectively. Of note, the sham/CIA group actually surpassed those exposed to specific oral pathobionts and revealed the highest expressions of IL-1β, IL-10, IL-6, IL-27, IL-17A, IFNβ and GM-CSF (Fig 5B).

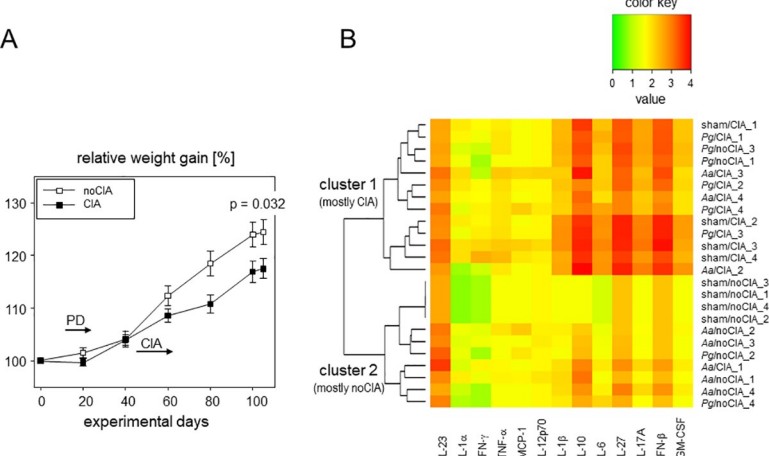

**Fig 5. CIA was associated with reduced weight gain and increased inflammatory cytokines.** (A) Weight curves show delayed weight gains in CIA mice. Symbols represent means ± SEM, numbers are n = 24 for CIA mice (n = 7 sham/CIA, n = 8 *Aa/CIA*, n = 9 *Pg*) and n = 23 for mice without CIA (n = 7 sham, n = 8 Aa, n = 8 Pg). P-values at endpoint result from two-tailed t-test. (B) The heat map summarizes the systemic expression levels of inflammatory cytokines (listed below the heat map) at endpoint. The color key is shown above, individual animals are listed on the right and the dendrogram on the left visualizes the connectedness between animals. N = 4 per group were analyzed.

## CIA did not exacerbate subsequent PD nor did PD impact on pre-exiting arthritis

In order to investigate whether CIA would impact on subsequent PD, we reversed the experimental setup and started out with the induction of arthritis before periodontal disease (Fig 6). Collagen type II priming was performed on experimental day, 1 followed by the boost on day 22. After the boost, mice were regularly scored for incidence and severity of arthritis until endpoint on day 105. Antibiotic treatment to facilitate subsequent colonization with the oral pathobionts was performed between experimental days 37 and 46, followed by the application of *Pg* and *Aa* from day 50 onwards until day 64. Application of sodium carboxy-methylcellulose in PBS served as sham treatment for PD induction. Note that the common endpoint on

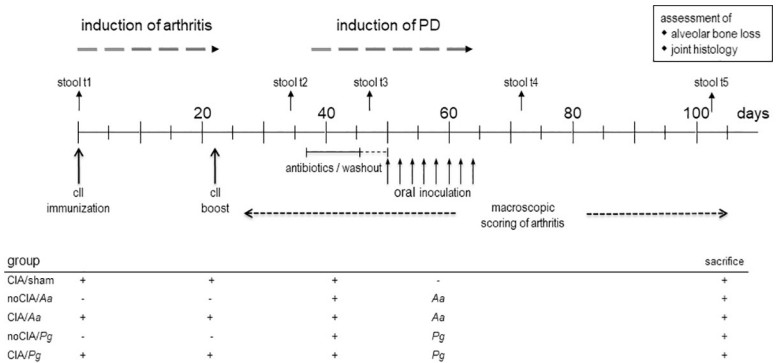

**Fig 6. The experimental setup of CIA induced before PD.** Periods for antibiotics treatment and macroscopic scoring of arthritis are indicated by horizontal lines and arrows, while individual days for immunization, boost, oral application of the pathobionts, as well as stool sampling are shown by vertical arrows. Experimental endpoint was day 105 and assessments of alveolar bone loss and joint histology were only made thereafter. Tables below the graphs summarize the individual experimental groups. *Pg*: *Porphyromonas gingivalis*; *Aa*: *Aggregatibacter actinomycetemcomitans*; sham: oral inoculation with PBS/ sodium carboxymethylcellulose.

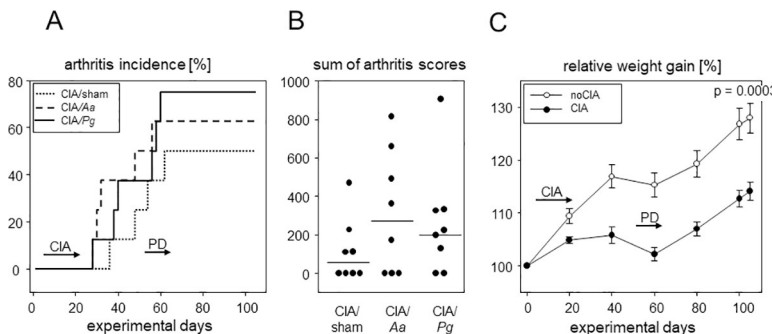

**Fig 7. PD did not exacerbate subsequent CIA.** (A) Inverted Kaplan-Meier curves show arthritis incidences over time. Horizontal arrows indicate periods of disease inductions. Numbers are n = 8 for all groups. (B) Dot plots summarize the sums of macroscopic arthritis scores at endpoint, assessed every other day starting from boost. (C) Weight curves show delayed weight gains in CIA mice. Symbols represent means ± SEM, numbers are n = 24 each for CIA and noCIA groups. P-value at endpoint resulted from Mann-Whitney test. Every dot represents a single mouse.

experimental day 105 implied more time to develop arthritis and less to develop alveolar bone loss compared to the setup in Fig 1.

Arthritis incidences were comparable between all experimental groups and at endpoint ranged between 50 (CIA/sham) and 75% (CIA/*Pg*) (Fig 7A). Subsequent PD induction had no additional impact. Likewise, accumulated arthritis scores at endpoint were comparable between sham and *Pg* or *Aa* treated groups (Fig 7B). Weight gain in the absence of CIA was constant except for the period of PD induction when oral inoculations demanded repeated anaesthesia. In contrast, mice induced for arthritis showed a delayed weight gain during the priming and boosting period and they remained at significantly reduced weights until endpoint (Fig 7C).

Induction of PD resulted in antibody responses towards the pathobionts that were comparable in the absence and presence of CIA and were also comparable to those observed in the reversed experimental setup (Fig 2D). Evaluating alveolar bone loss yielded significant increases in both, the *Aa* and the *Pg* groups which however, were again independent of CIA. Of note, the time frame to develop alveolar bone loss was only 50 days compared to 80 in the previous experimental setup, yet resulted in mean ABC-CEJ distances of 0.24 mm for *Aa* (0.21 mm in the previous setup) and 0.24 mm for *Pg* (0.22 mm in the previous setup), respectively (Fig 3C).

## *Pg* and *Aa* prompted specific signatures of synovial proteomes

To investigate whether inoculation with oral pathobionts influenced arthritis on a molecular level, we compared the synovial proteomes of the various experimental groups. To that extent, we isolated the synovial membranes of 20 knees, isolated the proteins and performed mass spectrometry. Of the 20 samples analyzed, 4 (2x CIA/Sham, 1x Sham/CIA, 1x *Aa*/CIA) had to be discarded due an overrepresentation of muscle and blood proteins which we attributed to flawed preparation. In the remaining batch, we identified 1275 different proteins or protein groups which were present in at least two samples and which contained at least two unique peptides. Of these proteins or protein groups, 24 were found to be differentially expressed depending on the experimental setup with a fold change > 2 (p < 0.05). Principal component analysis of the abundances of these proteins resulted in distinct clustering of the data (Fig 8A). The combination of oral inoculation with *Pg* and CIA for example–independent of the order of events—led to an up-regulation of the alpha and beta chains of the MHC class II (HB2 and HA2, Fig 8B). Likewise, the synovial membranes of arthritic mice inoculated with *Aa* exhibited

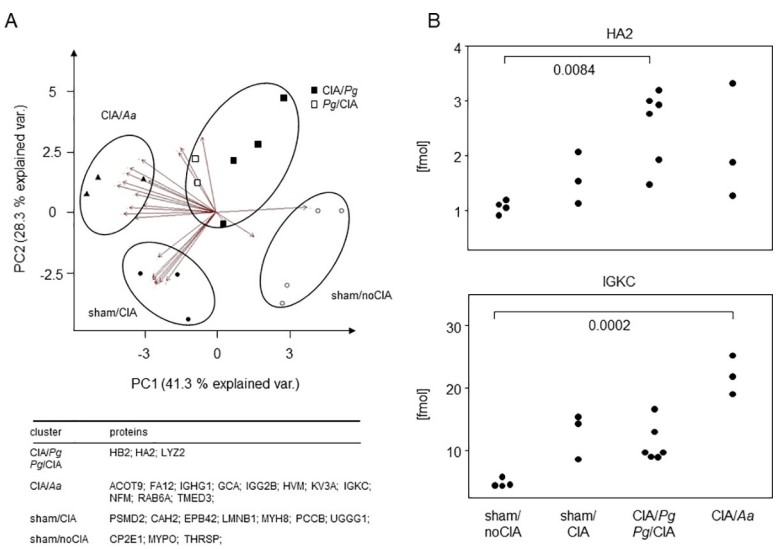

**Fig 8. Inoculation with oral pathobionts was reflected in the synovial proteome.** (A) PCA clustering based on protein abundances demonstrates that the synovial proteomes separated depending on arthritis induction and oral inoculation with either *Pg* or *Aa*. The table summarizes significantly upregulated proteins within each of the various clusters. (B) Dot plots exemplify quantitative expression data of MHC class II alpha (HA2) and immunoglobulin kappa light chains (IGKC). Asterisks denote statistically significant differences resulting from Kruskal-Wallis tests.

higher abundancies of immunoglobulin kappa light chains (IGKC, Fig 8B). However, not only proteins characteristic of an altered immune homeostasis were expressed differentially but also seemingly unrelated proteins like PSMD2, a member of the ubiquitin proteasome pathway. In summary both, arthritis by itself and in combination with oral pathobionts were differentially reflected in the synovial proteome.

## Discussion

We here used the mouse as a model to combine experimental arthritis with periodontal disease and we applied inverted orders of events to investigate the mutual impact both diseases exert on each other. The choice of DBA/1 x B10.Q F1 mice was based on their susceptibility for both, CIA and alveolar bone loss caused by orally inoculated bacteria [16, 17]. We successfully induced arthritis as macroscopically observed by redness and swelling of paws and randomly confirmed via histology showing cellular infiltrates as well as cartilage and bone erosions. Systemically we found elevated inflammatory cytokines and these findings are in line with previous descriptions of IL-1, IL-6 and TNFα being associated with bone erosion, joint destruction and increased disease activity [18, 19]. As for periodontal disease, we successfully induced alveolar bone loss however, did not detect signs of ongoing gingival inflammation at endpoint. This lack of histologically observed local inflammation is in line with results published for C57BL/6 yet contradicts findings in BALB/c mice [20, 21]. Whether indeed different mouse strains, different frequencies of oral inoculation, varying bacterial loads or time lapses between oral inoculation and evaluation of histological changes cause conflicting results remains to be determined. However, we did confirm successful inoculation with *Pg* and *Aa* twofold, via antibodies against the pathobionts and via systemic increases in inflammatory cytokines. Here, our data are in line with previous observations showing a raise in inflammatory cytokines after *Pg* and *Aa* exposure [22, 23].

As we did not assess oral dysbiosis in live animals around the time of exposure, we focused on the intestinal microbiome as a proxy, hoping to establish a non-invasive method to confirm

oral inoculation. Unfortunately, *Porphyromonadaceae* turned out to be common representatives of the murine stool and therefore, oral challenges with *Pg* or *Aa* did not result in any significant changes to the intestinal diversity [24, 25]. Therefore, intestinal diversity following oral inoculation may be restricted to the ileum, as suggested previously [20]. Alternatively, antibiotics treatment preceding inoculation with oral pathobionts may have exerted a lasting impact on the gut microbiome [26]. Likewise, antibiotics treatment may have blurred changes to the gut microbiome that have previously been shown to be associated with CIA commencement [26–28].

The picture emerging suggests that our mice were capable of clearing the oral infection and this has previously been confirmed for *Pg* [29]. Moreover, we observed less alveolar bone loss the longer the time lapse between oral inoculation and μCT assessment of the mandibles, further suggesting that at an early stage of periodontal disease, the murine alveolar bone possesses the capacity to remodel and regenerate. Even though the kinetics of clearing an oral infection and regenerating alveolar bone loss need to be analyzed in short term experiments, transient gingivitis and limited regain of lost alveolar tissue have already been described for humans [30, 31].

The reduced weight gain in mice suffering from CIA confirmed impaired health and suggested a major impact on the wellbeing of the mice. This did not apply to periodontal disease where a delay in weight gain could only be observed around the time of oral inoculation and was most likely attributable to repeated anesthesia. Unlike in our mice, previous studies in humans emphasized a causal relationship between periodontal infections and weight gain [32, 33]. However in humans, acceleration of periodontal diseases was correlated to obesity implying that either nutrition or elevated cytokines associated with obesity have an impact [34]. As in our experiments all mice received the same chow, we ruled out nutrition as an influence on weight gain. Instead, we believe that inflammatory cytokines which were strongly increased after inoculation with oral pathobionts impacted negatively on the weight.

Our main question–whether periodontal disease exacerbates RA and vice versa–requires a detailed response. While we did not see an impact of CIA on alveolar bone loss, this finding was independent of which disease was induced first. And even though our reversed experimental set up has not been described before, there is some literature presenting alveolar bone loss following induction of experimental arthritis [35, 36]. While we cannot fully explain these discrepancies, we can only speculate that the differences between mouse strains, between the modes to induce arthritis or the length of disease duration until mandibles were analyzed play an important role. Likewise, we did not observe a significant impact of periodontal disease on CIA. However, as indicated above, we believe that in our mice periodontal disease was transient and it is therefore difficult to evaluate if arthritis induction at the height of periodontal disease might turn out differently. Indeed, timing may play a role here as previous publications demonstrating an exacerbation of arthritis through periodontal disease opted for shorter time lapses between inductions of both diseases [37–39]. More differences may arise from the use of various mouse strains, different modes to induce arthritis to the point of collagen type II derived from chicken vs cow [29, 40]. In our experiments, the sums of arthritis scores at endpoint were higher for those mice that were subjected to CIA first however, we attribute this difference to different disease durations.

We ourselves have recently shown in a similar experimental setup, that pharmaceutical treatment of PD with metronidazole or chlorhexidine ameliorated periodontal disease and arthritis alike. Just as well, treatment of arthritis with Methotrexate reduced alveolar bone loss [16]. Even though these findings were suggestive of a linked pathogenesis between both diseases, the present results rather argue in favor of parallel disease developments.

Our most important data here is that even though oral inoculation with pathobionts did not result in the exacerbation of macroscopic arthritis or different histologies, *Pg* and *Aa*

prompted differential proteomic signatures. Indeed, this finding is corroborated by the work of Chukkapalli and colleagues who in a similar experimental setting demonstrated the homing of *Pg* to the inflamed joints [38]. Likewise, DNA from oral pathobionts has been identified in both, serum and synovial fluid of patients suffering from RA [41]. Our principal component analyses revealed that samples derived from mice that were orally inoculated with *Pg* and induced for arthritis featured elevated expression of MHC class II proteins. This finding is suggestive of an increased activity of innate antigen presenting cells and T lymphocytes. In contrast, oral inoculation with *Aa* in combination with arthritis led to increased expressions of immunoglobulin IgG1 and IgG2b as well as κ light chains, indicating a dominant humoral immune response. Our data thus describe differences on a molecular level that may simply result from different bacteria homing to the joints–but may also predict different disease entities. When transferred to the human situation, these different disease entities may predict differential responses to the various DMARDs available. While an entity characterized by elevated antigen presentation may respond better to TNFα blockade, a dominant humoral response may be attenuated by a B cell targeting therapy. Even though we have no explanation yet as to how oral pathobionts manipulate the synovial proteome on a mechanistic level, we consider it possible that periodontal disease shapes the subtype of arthritis and thereby sets the stage for later therapy response. A common response of arthritis and periodontal disease to antibiotic or antiseptic therapy is therefore gaining credibility. It will be interesting to dissect the window of opportunity for oral pathogens to impact on arthritis development and therapy response.

In summary, we here investigated the mutual impact periodontal disease and CIA exert on each other. While there was only little impact on the macroscopic level, the oral pathobionts *Pg* and *Aa* prompted differential proteomic signatures in the synovia suggesting a pathogen specific potential to shape individual disease subtypes.

## Methods and materials

### Mice

DBA/1 and B10.Q mice were originally purchased from Harlan Winkelmann (Borchen, Germany) and male F1 (DBA/1 x B10.Q) mice were bred in our animal care facility under specific pathogen free conditions, 12 hour light/dark cycles with food and water provided *ad libitum*. Mice entered into the experiments at an age of 8 to 12 weeks and were subsequently weighed and clinically scored every other day. Each experimental group consisted of 7 to 9 animals. The local state's animal care committee (Landesamt für Landwirtschast, Lebensmittelsicherheit und Fischerei Mecklenburg-Vorpommern) approved the experiments (7221.3-1-071/17) and all procedures were performed in agreement with the federal guidelines for animal experiments. All efforts were made to minimize suffering.

### Definition of a humane endpoint

During antibiotic treatment, inoculation of the bacteria and time after primary immunization and boost, animals were examined daily. In all other phases of the experiment—unless required otherwise by the condition of the animals—animals were examined every other day. Overall, no more than a moderate burden was tolerated, any higher exposure would have led to a humane endpoint for the respective animal. Criteria for humane endpoints were weight reduction of more than 20%, severe general condition, abnormal spontaneous behavior or open wounds located near the joints or paws. A total of 81 animals were included in our study of which 79 were euthanized on day 105 at the end of the experiment. Two were found dead after the second blood collection. All mice were anesthetized during oral inoculation, blood

sampling and prior to final cervical dislocation. After arthritis induction, the mice continuously received Tramadol via their drinking water.

## Bacterial strains

*Porphyromonas gingivalis* W83 strain was provided by the Institute of Medical Microbiology, Virology and Hygiene, University Medical Center Rostock, Rostock, Germany. Bacterial culture was performed under anaerobic conditions (10% $CO_2$, 10% $H_2$, 80% $N_2$) in Peptone-Yeast-Glucose (PYG) medium, which was supplemented with 5 µg/ml hemin and 1% vitamin K solution. *Aggregatibacter actinomycetemcomitans* (DSMZ, Braunschweig, Germany, DSMZ 11123) was cultivated at 37˚C under 5% $CO_2$-ambient atmosphere in brain heart infusion medium (BHI, Invitrogen, Carlsbad, CA, USA). Both species were grown over night and approximately 2 x $10^9$ colony-forming units (CFU) were washed with PBS and resuspended in 50 µL Dulbecco's Modified Eagle's Medium (DMEM). Aliquots were kept at −80˚C and CFUs were determined every six weeks to verify uniform viability. For inoculation, bacteria were thawed, pelleted and the supernatant removed.

## Anesthesia

For oral inoculation with periodontal pathogens, drawing blood from the orbital vein plexus and final sacrifice, mice were placed under deep anesthesia via intraperitoneal injections of 0.75 mg Esketamin (100 mg/ml, bela-pharm, Vechta, Germany) and 0.05 mg Xylazin (20 mg/ml, Bayer AG, Leverkusen, Germany) per 10 g of body weight. Hypothermia after oral inoculation and blood drawing was prevented by placing mice under an infrared lamp for at least 1 hour after narcosis induction.

## Collagen induced arthritis

CIA was induced as described previously [16]. In brief, mice were primary immunized via subcutaneous injection of 140 µg bovine type II collagen (Chondrex, Washington, USA) in 0.1 M acetic acid, emulsified in an equal volume of complete Freund's adjuvant (CFA, Becton, Dickinson and Company, Franklin Lakes, NJ, USA) at both sides of the base of the tail. Three weeks later, mice were boosted with 140 µg bovine type II collagen in 0.1 M acetic acid emulsified in an equal volume of incomplete Freund's adjuvant (IFA, Becton, Dickinson and Company, Franklin Lakes, NJ, USA).

## Induction of periodontal disease

Inoculation with oral pathobionts was performed as previously described [16]. In brief, mice initially received a 10 days course of antibiotics administered via the drinking water containing 2% of antibiotics (Cotrim K—ratiopharm 240 mg/5 ml, Ratiopharm, Ulm, Germany) followed by a 3 days washout period with tap water. For induction of periodontal disease, the prepared bacterial aliquots (2 × $10^9$ CFU) suspended in 50 µl PBS containing 2% sodium carboxy-methylcellulose (Sigma-Aldrich, St. Louis, MO, USA) were orally administered via pipette. The mixture was directly applied in 2 x 25 µl doses onto the right gingiva. Sham treated mice received 2% sodium carboxymethylcellulose in PBS. After inoculation, mice were denied food and drinking water for at least one hour. Inoculation with pathobionts was repeated every other day for a total of eight times.

## Blood samples

Blood samples were obtained 6 days after the last inoculation of bacteria, 10 days after the collagen boost and at the end of the experiment. Samples were left to clot at room temperature for

20 minutes and were then centrifuged at 1,500 rcf for 10 min. The serum was stored at -80˚C until downstream analyses were performed.

## Assessment of alveolar bone loss

Alveolar bone loss was assessed as previously described [15]. In brief, mandibles were fixed in 4% paraformaldehyde (PFA) for 7 days and then stored in 0.9% NaCl. Three-dimensional microcomputed tomography was performed using a SkyScan 1076 micro-CT scanner (Bruker, Billerica, MA, USA). For reproducible evaluations, 18 measuring points were defined for each right hemi-mandible: after three-dimensional alignment, 3 geometrically distributed sites were determined for the lingual and buccal side of each tooth. Distances from the cemento-enamel junction (CEJ) to the alveolar bone crest (ABC) were then measured in millimeters and in parallel for all groups.

## Macroscopic scoring of arthritis and histological examination

Onset and severity of arthritis were scored as previously described [15]. In brief, scoring was performed every other day following the boost. Macroscopic signs of arthritis including swelling and erythema were rated with 5 points each for an affected paw or wrist joint, and with 1 point for each affected digit, respectively. A maximum score of 15 points per hind and a maximum score of 14 for each fore paw and 58 points per mouse could thus be reached. Arthritis incidence was determined at a clinical score of at least 1. After sacrifice, paws were collected and fixed in 4% formalin for two weeks. After washing the paws with tap water for 30 minutes, they were decalcified using Usedecalc (Medite, Burgdorf, Germany) and embedded into paraffin. Tissue slices were stained with hematoxylin and eosin (HE) and microscopic images were obtained using 12.5 and 200 fold magnifications (Axioplan 2, Zeiss, Oberkochen, Germany).

## Microbiome analysis

Microbiome analyses were performed as previously described [15]. In Brief, fresh stool samples were collected from individual mice on the days indicated and were stored at -80˚C until use. For further analysis, stool was thawed, homogenized via Fastprep-24 (MP Biomedicals, Santa Ana, CA, USA) with 6 m/s for 1 min using the ZR-96 BashingBead Lysis Tubes (Zymo Research, Irvine, CA, USA) and DNA was eluted using the ZymoBIOMICS DNA Miniprep Kit (Zymo Research, Irvine, CA, USA) following the manufacturer's instructions. DNA concentration were measured via NanoDrop 2000 spectrophotometer (Thermo Fisher Scientific, Waltham, MA, USA).

**16SrRNA PCR.** Amplicon PCR was performed with microbial genomic DNA using concentrations of 5 ng/μl in 10 mM Tris pH 8.5. PCR amplification of the V3/V4 regions of bacterial 16S rRNA encoding gene was carried out using the primers Pro341-XT (`TCG-TCG-GCA-GCG-TCA-GAT-GTG-TAT-AAG-AGA-CAG-CCT-ACG-GGN-BGC-ASC-AG`) and Pro805-XT (`GTC-TCG-TGG-GCT-CGG-AGA-TGT-GTA-TAA-GAG-ACA-GGA-C-TA-CNV-GGG-TAT-CTA-ATC-C`) which resulted in amplicon sizes smaller than 550bp. The details of library constructions, such as Index PCR, PCR clean-up 2, library quantification, normalization and pooling were performed corresponding to the Illumina "16S Metagenomic Sequencing Library Preparation" protocol.

Bioanalyzer DNA 1000 chips (Agilent Technologies, Santa Clara, CA, USA) and Qubit kits (Thermo Fischer Scientific, Waltham, MA, USA) were applied for quantity and quality controls of each individual sample library and the final library pool. 5 pM of the final library mixture, containing at least five percent PhiX control, were exposed to one individual sequencing run using a 2x250 or 2x300 cycle 3 reagent cartridge on an Illumina MiSeq machine. All raw

data fastq files were used for sequence data analyses. The raw sequencing fastq files will be submitted to the BioProject database.

**Sequence data analysis.** Quality filtering (permitted length: 440–466 bp, no ambiguous bases allowed), merging of duplicated sequences and alignment to the reference database (https://www.mothur.org/wiki/Silva_reference_files#Release_128) were done using Mothur [42]. Only OTUs with total abundance > = 3 were considered. Sequences from archaea, chloroplasts, eukaryota and mitochondria were removed. For descriptive community analysis and PCA plots, the CRAN package 'vegan' has been used (https://cran.r-project.org/web/packages/vegan/index.html). Similarity was calculated as Jaccard index, as a measure for dissimilarity Bray-Curtis has been used. For statistical analysis of differences between read counts at different time points, Kruskal-Wallis tests with pairwise multiple comparisons by Nemenyi-tests were used. The raw sequencing fastq files were submitted to and are available at the BioProject database under the ID "PRJNA663999"and "PRJNA664015", respectively.

## Serum analysis

Evaluation of IgG antibodies against *Pg* and *Aa* was performed as previously described [15]. For coating of *Pg* proteins onto ELISA plates, $2 \times 10^9$ cfu were resuspended in 600 μL $CO_3^{2-}$/ $HCO_3^-$ buffer (pH 9.4) containing protease inhibitor cocktail (Roche/Sigma Nr: 04693159001) and 1 mM EDTA, before homogenization at 6000 rpm, 3 times for 30 s using Precellys 24 (Stretton Scientifc, Stretton, UK). After performing a Bradford assay for protein content quantification, 100μL of a 1 μg/mL protein solution was coated overnight onto MediSorp ELISA plates (Thermo Fisher Scientific, Waltham, MA, USA). Plates were washed with PBS-Tween 20 (0.05%) and blocked with 2% bovine serum albumin (Sigma-Aldrich, St. Louis, MO, USA). Mouse serum was applied at dilutions of 1:200. After 1.5 h at RT, plates were washed 3 times and incubated with detection antibody (STAR13B (Rb F(ab')2 a-mu IgG:HRP)) at a dilution of 1:1000 for 1 h. For color reaction, 100μL TMB substrate (Biolegend, San Diego, CA, USA) were added. Optical density was determined at an absorbance of 450 nm using an automated plate reader (Anthos htIII; Anthos Labtec Instruments, Salzburg, Austria). For determination of serum cytokine levels a LEGENDplex mouse inflammation panel assay (BioLegend, San Diego, CA, USA) was used. Flow cytometric measurement was performed on a FACSVerse (Becton, Dickinson and Company, Franklin Lakes, NJ, USA).

## Analysis of synovial membranes

**Synovial membrane isolation.** For collection of synovial tissue, we followed the protocol of Valverde-Franco et al. [43] with few modifications. A surgical microscope (M715, Leica, Wetzlar, Germany) and number 10 scalpel blades (Dahlhausen, Cologne, Germany) were used. After sacrifice, mice were placed dorsally and a medial arthrotomy from the tibial plateau to the proximal patella was conducted. The patellar tendon was intersected, a lateral arthrotomy performed and the patella folded up to expose and isolate synovial membrane and joint capsule. A u-shaped incision was performed around the trochlea femoralis and the separated synovial membrane was grabbed with forceps and was excised. The synovial tissue was shock frozen in liquid nitrogen and then stored at -80˚C. Analysis of synovial membrane.

**Sample preparation for whole protein analysis.** For protein extraction, 20 μL of lysis buffer (2% sodium dodecyl sulfate, 50 mM dithiothreitol, 50 mM Tris/HCl pH 7.6) were applied to the synovial membranes. After 5 min incubation at 95˚C, extraction was completed via 5 min of ultrasonication using a bath sonicator. Quantification of protein contents was performed using the Qubit Protein Assay Kit (Q33211, Thermo Fisher Scientific, Waltham, MA, USA) according to the manufacturer's instructions with subsequent measurement on a Qubit

Fluorometer (Thermo Fisher Scientific, Waltham, MA, USA). For the filter aided sample preparation (FASP), samples were mixed with 200 μL solution UA (8 M urea, 100 mM Tris/HCl pH 8.5) as previously described [44]. Sample solutions were applied and filters were centrifuged for 15 min. After washing with 100 μL UA, 50 μL of 50 mM iodoacetamide in UA was added followed by 20 min incubation in the dark for complete alkylation of cysteine residues. Filters were washed twice with 100 μL or 50 μL UA followed by three subsequent washing steps with 100 μL, 75 μL or 50 μL of 50 mM ammonium bicarbonate. On-filter digestion was performed using 1 μg trypsin (V5111, Promega, Madison, WI, USA) in 40 μl of 50 mM ammonium bicarbonate for each 50 μg of sample protein with over-night incubation at 37°C in a wet chamber. Peptides were collected by centrifugation and fresh trypsin solution was added onto the filter for a second digestion for 90 min. After centrifugation, the combined digests were acidified with trifluoroacetic acid (final concentration 0.25%), concentrated by use of a centrifugal evaporator and diluted to a final volume of 40 μl with a solution containing 2% acetonitrile and 0.1% formic acid in water. Thereafter, quantification of peptide contents were performed on a Qubit Fluorometer as described above.

**Analysis by nanoLC-HDMS$^E$.** Separation and mass spectrometric analysis of peptides were performed as described previously on a nanoAcquity UPLC system (Waters, Manchester, UK) coupled to a Waters Synapt G2-S mass spectrometer [45]. In short, 160 ng of sample peptides supplemented with 40 fmol of Hi3 ClpB_ECOLI standard for protein absolute quantification (Waters, Manchester, UK) were trapped and desalted using a pre-column (ACQUITY UPLC Symmetry C18, 5 μm, 180 μm x 20 mm, Waters, Manchester, UK) at a flow rate of 10 μl/min for 4 min with 99.9% mobile phase A (0.1% formic acid in water) Subsequently, peptides were separated on an analytical column (ACQUITY UPLC HSS T3, 1.8 μm, 75 μm x 250 mm, Waters, Manchester, UK) using a gradient from 3% to 32% mobile phase B (0.1% formic acid in acetonitrile) over 120 min at a flow rate of 300 nL/min and 55°C. Data acquisition for peptide ions after ion-mobility separation was conducted with data-independent alternations of scans at low and elevated collision energy (CE) to obtain precursor and fragment ions for 0.6 s respectively (referred to as HDMS$^E$).

**Proteomics data processing, protein annotation and quantification.** Progenesis QI for Proteomics version 4.1 (Nonlinear Dynamics, Newcastle upon Tyne, UK) was used for raw data processing, protein identification and quantification. Alignment compensation of the HDMS$^E$ data was conducted to account for variations of LC separation in-between runs. Peptide and protein identifications were obtained by searching against a Mus musculus database containing 17,006 reviewed protein sequences (UniProt release 2019_02) appended with the sequences of ClpB_ECOLI (P63284) and porcine trypsin. Two missing cleavage sites were allowed, oxidation of methionine residues was considered as variable modification, and carbamidomethylation of cysteines as fixed modification. The false discovery rate based on the simultaneous search of a decoy database was set to 1%. Peptides were required to be identified by at least three fragment ions and proteins by at least six fragment ions from minimally two peptides. Subsequently peptide ion data were filtered to retain only peptide ions that met the following criteria: (i) identified at least two times within the dataset, (ii) ion score greater 5.5, (iii) mass error below 13.0 ppm, (iiii) at least 6 amino acid residues in length. Finally, only proteins with at least two unique peptides were included into the quantitative analysis. For label-free quantification, the Hi3 method implemented into the Progenesis QI for Proteomics workflow was applied using the Hi3 ClpB_ECOLI standard (Waters, Manchester, UK) as a reference [46]. Hi3 peptide quantification uses the sum of the signal intensities of the three most abundant peptides of each protein, divided by the sum of the signal intensities of the three most abundant peptides of the internal standard, multiplied by the amount of standard applied to the column. Downstream multivariate and single variate analyses were performed on

protein abundances that were significantly different between all samples (see section below). Data are available via ProteomeXchange with identifier PXD020397.

## Statistical analysis

Statistical analysis was executed by using Sigma Plot (Version 13.0, Systat Sostware, Erkrath, Germany) or SPSS (Version 25, IBM, Armonk, NY, USA). Data were tested for Gaussian distribution by dint of Shapiro-Wilk-test. For comparisons of 2 groups, two-tailed t-tests were performed for data following Gaussian distribution. Otherwise, Mann-Whitney-U-test were applied. For the comparison of more than two groups, ANOVA (in case of Gaussian distribution) or Kruskal-Wallis-tests with post-hoc tests were utilized. To represent incidence of collagen induced arthritis, Kaplan-Meier-Estimator was applied and analyzed by using Log Rank-, Breslow- and Tarone-Ware-tests. Multivariate analyses of proteome data were performed in R as described [47]. Clustering of sample data was computed using normal ellipses with a confidence interval of 0.95. Single variate analysis was performed using Kruskall-Wallis one-way analysis of variance with posthoc Mann-Whintey-U test and Bonferroni-Holm correction for multiple comparisons. The alpha levels for all tests were 0.05.

## Supporting information

**S1 File.**
(PDF)

## Acknowledgments

The authors would like to acknowledge the support of Jana Bull for supply of bacteria and Paul Lübcke, Meinolf Ebbers, Katharina Wenndorf, Michael Müller and Wendy Bergmann for their help with animal handling, μCT measurements and evaluation as well as flow cytometry. The authors would also like to thank Michael Sämann for help with 3D reconstructing the mandibles.

## Author Contributions

**Conceptualization:** Anna-Lena Buschhart, Lennart Bolten, Bernd Kreikemeyer, Brigitte Müller-Hilke.

**Data curation:** Anna-Lena Buschhart, Lennart Bolten, Johann Volzke, Katharina Ekat, Susanne Kneitz, Stefan Mikkat.

**Formal analysis:** Anna-Lena Buschhart, Lennart Bolten, Johann Volzke, Susanne Kneitz, Stefan Mikkat.

**Investigation:** Anna-Lena Buschhart, Lennart Bolten, Susanne Kneitz.

**Methodology:** Anna-Lena Buschhart, Lennart Bolten, Susanne Kneitz.

**Project administration:** Brigitte Müller-Hilke.

**Resources:** Brigitte Müller-Hilke.

**Software:** Susanne Kneitz, Stefan Mikkat.

**Supervision:** Brigitte Müller-Hilke.

**Validation:** Anna-Lena Buschhart, Lennart Bolten, Johann Volzke, Susanne Kneitz, Stefan Mikkat, Brigitte Müller-Hilke.

**Visualization:** Anna-Lena Buschhart, Lennart Bolten, Johann Volzke, Brigitte Müller-Hilke.

**Writing – original draft:** Anna-Lena Buschhart, Lennart Bolten, Brigitte Müller-Hilke.

**Writing – review & editing:** Anna-Lena Buschhart, Lennart Bolten, Johann Volzke, Katharina Ekat, Susanne Kneitz, Stefan Mikkat, Bernd Kreikemeyer, Brigitte Müller-Hilke.

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
