## [Decision Letter · Decision Letter 0]

30 Sep 2020

PONE-D-20-26647

Periodontal pathogens alter the synovial proteome

Periodontal pathogens do not exacerbate macroscopic arthritis but alter the synovial proteome in mice

PLOS ONE

Dear Dr. Müller-Hilke,

Thank you for submitting your manuscript to PLOS ONE. After careful consideration, we feel that it has merit but does not fully meet PLOS ONE’s publication criteria as it currently stands. Therefore, we invite you to submit a revised version of the manuscript that addresses the points raised during the review process.

We look forward to receiving your revised manuscript.

Kind regards,

David Douglass Brand

Academic Editor

PLOS ONE

Journal Requirements:

2. Please download the ARRIVE Guidelines 2.0, "Essential 10" checklist and submit it with your revision. You may find it here: https://arriveguidelines.org/resources/author-checklists.

3.Thank you for stating the following in the Acknowledgments Section of your manuscript:

[Purchase of the

Illumina MiSeq was kindly supported by the EU-EFRE (European Funds for Regional Development)

program and funds from the University Medicine Rostock awarded to BK.]

 [The author(s) received no specific funding for this work.]

Additional Editor Comments (if provided):

In reviewing this manuscript, both reviewers have suggested that a significant revision is in order before this work is ready for publication. Each reviewer has provided many different areas that they have suggested will strengthen the work. As academic editor, I would mirror the comments provided by reviewer 1 at the conclusion of the review. While it is very important to report all findings in a study and to draw the best conclusions possible from those findings, if some of those findings are in direct contrast to previously published works (as is the case here) it is important to use the discussion section to compare and contrast the methods, reagents and anything else that might differ between the studies so that the appropriate conclusions can be drawn. It in no way means that one study is right and the other is wrong, it's just that the interpretations should include a careful consideration of all existing data.

There is clearly a very complex relationship between periodontal disease and arthritis, and this complexity is compounded by the use of rodent models to allow us to dissect some of these relationships and mechanisms. This makes it imperative to have multiple perspectives from different laboratories approaching the problem. I would encourage you to take these reviews into consideration as you revise this important work.

Reviewers' comments:

Reviewer's Responses to Questions

**Comments to the Author**

1. Is the manuscript technically sound, and do the data support the conclusions?

Reviewer #1: Partly

Reviewer #2: No

2. Has the statistical analysis been performed appropriately and rigorously? 

Reviewer #1: Yes

Reviewer #2: No

3. Have the authors made all data underlying the findings in their manuscript fully available?

Reviewer #1: Yes

Reviewer #2: Yes

4. Is the manuscript presented in an intelligible fashion and written in standard English?

Reviewer #1: Yes

Reviewer #2: No

5. Review Comments to the Author

Reviewer #1: The manuscript “Periodontal pathogens do not exacerbate macroscopic arthritis but alter the synovial proteome in mice” aims to analyze the synovial proteome by using a model of arthritis and oral infection with two periodontopathogenic bacteria. Overall, the manuscript is well-done and results bring novelty to the literature, however, there are some concerns that need be addressed.

--The major point refers to the experimental design. While it is quite interesting, the accurately comparison between all groups is limited. The authors used two distinct models, one with PD induced first CIA and other with PD induced after CIA. In both situations, significant changes in oral microbiome are expected. What remains undetermined is the oral load of Aa and Pg in these situations. Were the differences between Aa and Pg CIA groups a result of the capacity of oral colonization of these bacteria? It is pivotal to link findings to systemic and joint inflammation. Moreover, the bacterial translocation is an event reported in the course of arthritis and periodontal disease, and this would account for differences in synovia proteome. Authors should address these points to support the hypothesis.

--The objectives of study are not clearly outlined in abstract and introduction.

--Introduction should focus in the main subject of study.

--While the experimental groups are clearly represented in Fig 1 the, the parameters of these groups were not consistently presented in the results section as pointed bellow. The subheadings of results section also require revision. A standardization of group’s nomenclature throughout the manuscript is necessary.

--Fig 2A, authors show the anti-Aa and anti-Pg ab production, in animals inoculated with respective bacteria. Did the authors perform this measurement in the course of CIA? It would be interesting to demonstrate the dynamic of Ab production in CIA/PD or PD/CIA groups. The justification that CIA mice were not included because of possible interferences of immunization and boost in the presence of Freund´s adjuvant is not justified since appropriated controls were included in all experiments to solve these problems.

--Fig 2C, please show microscopic images for Aa group. Importantly, please present data regarding alveolar bone loss for all groups.

--Line 133. Authors state that “Unfortunately, the minute anatomy of the murine oral cavity did not allow for the assessment of acute gingivitis and periodontitis in live animals. We therefore concentrated on antibodies against oral pathobionts, systemic cytokines and the gastrointestinal microbiome as proxies for successful colonization in the course of the experiments and evaluated gingival histology and alveolar bone loss at endpoint.” Please re-state it because these parameters may be assessed in periodontal tissues by a number of methods including histology, immunohistochemistry, biochemical assays (e.g. ELISA, MPO) and others and some of these were used in the study.

--Line 149-150. Please specify what means “neither signs of inflammation nor bacterial nests containing Pg or Aa…” since only H&E stained histology sections, without using specific methods (e.g. immunostaining), do not allow the identification of these bacteria in periodontium.

--Please describe variations in systemic cytokine responses presented in Fig. 2B.

--Fig 3A. MicroCT images should be presented for all groups. Also, Indicate if sham is sham/CIA or sham/no CIA and the same for Pg.

--Fig 4A. Given that synovia is one of the main focus of study, it is important to show histology for all groups.

--Fig 4C in the left of heat map there is an identification of animals as Pg/noCIA (on lines 3 and 4) but on right column these animals are identified in the CIA group. Please check it. Also, if possible reorganize the order of animals to facilitate the data interpretation.

--Fig 5B. In the groups in which PD was induced before arthritis, the time of second boost until the euthanasia was significantly lower compared to animals with PD induced after arthritis. Thus, it is expected that sham/CIA had lower arthritis incidence compared to CIA/sham. However the result is the opposite, about 70% for sham/CIA and 50% for CIA/sham. In the fig 5C the authors demonstrate that score for sham/CIA is lower than CIA/sham, what is coherent but not solve the apparent inconsistency seen in Fig 5B.

--Fig 6A. Why no data from animals without CIA but with Pg and Aa oral inoculation were included? Were the CIA/Pg and Pg/CIA presented as one? These findings are important to draw the conclusion.

--Data from the present study show important differences when compared with previous publications. For example, CIA groups exhibited almost no alveolar bone loss; few changes in joints were seen when Pg and Aa infection were performed before CIA. In contrast, previous studies account that arthritis per si induces alveolar bone loss and that Pg inoculation increases joint damage in CIA animals. Also, the lack of periodontal inflammation in Aa and Pg groups is dissimilar from previous literature. These differences should be discussed in details to substantiate further studies.

Reviewer #2: Buschhart et al examined the relationship between periodontal disease and inflammatory arthritis in two experimental protocols, in one the periodontal disease was induced prior to induction of arthritis, and in the other arthritis was induced prior to periodontal disease. The authors examined the outcome of the two diseases in these two experimental protocols.

The general take-home message is that, in these experimental set-ups, there was no effect of one disease on the other. These are still important findings to publish, however there are major concerns with the current version of the manuscript.

This reviewer found the manuscript very difficult to follow, and that is the gist of many of the comments below.

The other main problem is one of over-interpretation. Small differences that do not reach significance should not be presented as findings. This is true for the “slight” and “minor” changes found throughout – the major finding of the manuscript is that the diseases do not impact upon each other in mice under the experimental protocols presented.

Unfortunately, the difference in synovial proteome touted in the title and abstract, lead the reader to expect a more robust finding. In fact, the proteomic analysis was conducted on a small subset of the experimental groups, and the significance shown is a comparison to sham treated mice (no CIA, no pd) rather than to mice that did not receive the bacteria but were immunized to collagen. Therefore, here too the interpretation is not justified. This is especially true given the important negative finding that pd does not worsen arthritis.

Specific comments:

The authors performed two independent experimental protocols, and show that nicely in figure 1. These protocols address independent questions! After that it is confusing whether we are being shown data from 1A or 1B or both – this has to be made very clear and it’s too confusing to show all joined together such as in 3B, especially since other figures like figure 4 are showing only the protocol of 1A. then figure 5 shows data for both protocols. It’s just too difficult to follow in this reviewer’s opinion. Best would be to show 1A and all the experimental evaluations of 1A, then show 1B and the evaluations of 1B. If there are comparisons between 1A and 1B (again these are independent questions, and performed independently), then that would come afterwards.

Hard to follow the numbers in each group, and what was tested for each group. For example, figure 2A shows only sham vs. bacteria and the text provides the n of each group (large numbers for each group). Do these groups include +/-CIA? Figure 2B and 4C show heat maps but it’s unclear if the cytokine arrays were performed on all mice or only a select few mice per group. What was the actual group size for each endpoint measured? The heat maps also list individual mice not according to their group, further complicating the presentation. Some statistical comparison between groups is necessary to justify the conclusions that the authors present in the results section about induction or lack of induction of systemic inflammation.

Line 135: It is unclear why the authors chose to analyze the gastrointestinal microbiome as a “proxy” for colonization of the oral cavity. They could have tested oral dysbiosis induced by Pg or Aa lavage – it has been shown by multiple groups that Pg induces an increase in the total anaerobic bacterial counts in the oral cavity detected by swabbing (no need to sacrifice the animals). Alternatively, they could have analyzed colonization of the oral cavity with the strains of bacteria they administered. Is there a reference for using the feces microbiome as a proxy for Pg or Aa colonization of the oral cavity? According to their results the gut microbiome did not reflect their intervention so what was learned? It’s confusing and misleading as presented. The suggestion that bacteria induced a “minor” change in the gut microbiome, without showing statistical significance, is not justified. Furthermore, despite showing no dysbiosis induced by CIA, in the discussion line 299 the result is presented as if CIA indeed induced perturbation of the gut microbiome.

Line 185 reads “Figures 4A and B show macroscopic and histologic pictures of arthritic paws in mice that were subjected to periodontal disease induction before CIA” – actually what seems to be shown is one representative mouse of the control (no pd, no cia) group vs. one representative mouse of the CIA group (no pd). Therefore, the line is incorrect. In fact, what needs to be shown is the scoring of the CIA in all groups (caliper measurements, incidence and severity).

Figure 3b should compare independently the CIA+bacteria group to the CIA-bacteria, and the CMC-CIA to bacteria-CIA. Presently the comparison groups together the +/-CIA groups in the sham vs Pg or Aas with and without CIA.

Minor:

The legends appear in the text rather than at the end.

Line 395-400: clarify the method – are you administering the bacteria by gavage (in which case 25 ul does not seem realistic, and no gavage needle size is mentioned), or did you “apply” the bacteria? If applied, clarify what you mean.

Line 401 – what is “initial” about this?

Line 112 – as far as I understand it has not been demonstrated that systemic abx are necessary and indeed “facilitate colonization.” If you have a reference for this, please include. If not, please omit the claim. My understanding is that the model employs systemic abx based on the hypothesis that abx will facilitate colonization but that when it was tested the abx were shown not to be necessary.

Line 186 – before jumping to cytokines the authors should say something about the arthritis incidence and scores – these should be shown.

Last line of introduction – change synoviale to synovial

Line 412 “tree” should be “three”

6. PLOS authors have the option to publish the peer review history of their article (what does this mean?). If published, this will include your full peer review and any attached files.

Reviewer #1: No

Reviewer #2: No

---

## [Author Response · Author response to Decision Letter 0]

6 Nov 2020

PONE-D-20-26647

Dear Mr. Brand,

We are grateful for the opportunity to resubmit our revised manuscript and we appreciate the reviewers’ comments as they pointed out misunderstandings and weaknesses - and therefore helped to improve the manuscript.

Please, find below our point-by-point reply. We resubmit our manuscript in two versions, 

i) containing all changes marked in red and green in response to Reviewer 1 and 2 respectively, and 

ii) all black, only. 

Reviewer 2 suggested major (but helpful) changes concerning the order of results and figures, so that we now report first on all results concerning the experimental setup including periodontal disease induced before arthritis and then report on the reversed experimental setup, followed by the proteome analyses at the end. As a consequence, the whole manuscript was revised – leaving no line numbering and hardly any figure the same as before.

moreover, we 

- revised our styles according to the PLOS ONE’s style requirements, 

- submit ARRIVE guidelines and

- would like to move our statement about the EU-EFRE support to the Funding Statement section of the online submission form

 We hope that our revision addressed all concerns raised and look forward to a positive review.

Sincerely

Brigitte Müller-Hilke

Editor Comments:

In reviewing this manuscript, both reviewers have suggested that a significant revision is in order before this work is ready for publication. Each reviewer has provided many different areas that they have suggested will strengthen the work. As academic editor, I would mirror the comments provided by reviewer 1 at the conclusion of the review. While it is very important to report all findings in a study and to draw the best conclusions possible from those findings, if some of those findings are in direct contrast to previously published works (as is the case here) it is important to use the discussion section to compare and contrast the methods, reagents and anything else that might differ between the studies so that the appropriate conclusions can be drawn. It in no way means that one study is right and the other is wrong, it's just that the interpretations should include a careful consideration of all existing data.

There is clearly a very complex relationship between periodontal disease and arthritis, and this complexity is compounded by the use of rodent models to allow us to dissect some of these relationships and mechanisms. This makes it imperative to have multiple perspectives from different laboratories approaching the problem. I would encourage you to take these reviews into consideration as you revise this important work.

you hit a weak spot here. We gladly refer to the respective literature and discuss explanations for conflicting results (see LL 319 -332)

Reviewer #1: The manuscript “Periodontal pathogens do not exacerbate macroscopic arthritis but alter the synovial proteome in mice” aims to analyze the synovial proteome by using a model of arthritis and oral infection with two periodontopathogenic bacteria. Overall, the manuscript is well-done and results bring novelty to the literature, however, there are some concerns that need be addressed.

--The major point refers to the experimental design. While it is quite interesting, the accurately comparison between all groups is limited. The authors used two distinct models, one with PD induced first CIA and other with PD induced after CIA. In both situations, significant changes in oral microbiome are expected. What remains undetermined is the oral load of Aa and Pg in these situations. Were the differences between Aa and Pg CIA groups a result of the capacity of oral colonization of these bacteria? It is pivotal to link findings to systemic and joint inflammation. 

We are aware of the limitation of not having assessed the oral loads of Pg and Aa. However, we simply did not have permission for an additional fixation of the animals or even additional anesthesia, in order to perform these oral swabs. We will though in future experiments request more mice in order to sacrifice them at an early stage of periodontal disease, analyze the oral microbiome and perform early histology.

Moreover, the bacterial translocation is an event reported in the course of arthritis and periodontal disease, and this would account for differences in synovia proteome. Authors should address these points to support the hypothesis.

Done, please see discussion, L 342/343

--The objectives of study are not clearly outlined in abstract and introduction.

Revised, please see L 67-71

--Introduction should focus in the main subject of study.

Revised, please see L 47-54 and 59-62

--While the experimental groups are clearly represented in Fig 1 the, the parameters of these groups were not consistently presented in the results section as pointed bellow. The subheadings of results section also require revision. A standardization of group’s nomenclature throughout the manuscript is necessary.

done

--Fig 2A, authors show the anti-Aa and anti-Pg ab production, in animals inoculated with respective bacteria. Did the authors perform this measurement in the course of CIA? It would be interesting to demonstrate the dynamic of Ab production in CIA/PD or PD/CIA groups. The justification that CIA mice were not included because of possible interferences of immunization and boost in the presence of Freund´s adjuvant is not justified since appropriated controls were included in all experiments to solve these problems.

We only measured the presence of antibodies against Pg and Aa at endpoint. However, the respective Figures (now 3B and 3C) have been revised to the extent that all experimental groups are shown individually. Maybe that in itself addresses some of your concern/question. As for the comment “not including CIA mice” – this referred to the cytokine measurements and in order to avoid any misunderstanding, we rephrased. Please, see L 104-106

--Fig 2C, please show microscopic images for Aa group. Importantly, please present data regarding alveolar bone loss for all groups.

Done (now Figure 3A)

--Line 133. Authors state that “Unfortunately, the minute anatomy of the murine oral cavity did not allow for the assessment of acute gingivitis and periodontitis in live animals. We therefore concentrated on antibodies against oral pathobionts, systemic cytokines and the gastrointestinal microbiome as proxies for successful colonization in the course of the experiments and evaluated gingival histology and alveolar bone loss at endpoint.” Please re-state it because these parameters may be assessed in periodontal tissues by a number of methods including histology, immunohistochemistry, biochemical assays (e.g. ELISA, MPO) and others and some of these were used in the study.

As stated above, we will in future experiments investigate the oral microbiome at the early stages after inoculation. However, histology and immunohistochemistry will still require the animal to be sacrificed - and we simply did not include sufficiently enough animals to sacrifice some at an early stage and let others survive to evaluate alveolar bone loss at an advanced stage.

--Line 149-150. Please specify what means “neither signs of inflammation nor bacterial nests containing Pg or Aa…” since only H&E stained histology sections, without using specific methods (e.g. immunostaining), do not allow the identification of these bacteria in periodontium.

Agreed – and revised, please see L 109/110

--Please describe variations in systemic cytokine responses presented in Fig. 2B.

done, please see L 101-104

--Fig 3A. MicroCT images should be presented for all groups. 

We understand the intention behind this comment – and even though we introduced another micro-CT image, we dare to disagree. Representative images (of ten groups!) would suggest that there is no difference between the sham/noCIA and sham/CIA groups. However, the microCT pictures merely are supposed to illustrate how we performed the measurements – the summaries shown in (now) 3B and 3C are much more informative as they allow statistics to be performed.

Also, Indicate if sham is sham/CIA or sham/no CIA and the same for Pg.

done

--Fig 4A. Given that synovia is one of the main focus of study, it is important to show histology for all groups.

Well – the same applies here as with the comment to Fig 3. Inflammation resulting from CIA cannot be distinguished between the various experimental groups on a microscopic level via histology. Also, scoring of arthritis is not performed on a microscopic but on the macroscopic level. Therefore, the macroscopically determined scores provide more information and allow statistical analyses to be performed.

--Fig 4C in the left of heat map there is an identification of animals as Pg/noCIA (on lines 3 and 4) but on right column these animals are identified in the CIA group. Please check it. Also, if possible reorganize the order of animals to facilitate the data interpretation.

Agreed and revised into “cluster 1 – mostly CIA” and “cluster 2 – mostly noCIA”. However, Supervised ordering of samples within an hierarchically clustered heatmap would be most unusual as it interferes with the unsupervised data mining and even renders it obsolete.

--Fig 5B. In the groups in which PD was induced before arthritis, the time of second boost until the euthanasia was significantly lower compared to animals with PD induced after arthritis. Thus, it is expected that sham/CIA had lower arthritis incidence compared to CIA/sham. However the result is the opposite, about 70% for sham/CIA and 50% for CIA/sham. In the fig 5C the authors demonstrate that score for sham/CIA is lower than CIA/sham, what is coherent but not solve the apparent inconsistency seen in Fig 5B.

Yes – and no. We define onset of arthritis as the day on which the first swollen joint is observed – which usually is between day 6 and 40 following the boost. Even in the setup with PD induced before CIA, mice sat for 41 days after the boost (before they were sacrificed). We therefore did not expect different incidences between both experimental setups. And indeed, none of the differences observe were statistically significant. 

--Fig 6A. Why no data from animals without CIA but with Pg and Aa oral inoculation were included? Were the CIA/Pg and Pg/CIA presented as one? These findings are important to draw the conclusion.

Again, an interesting point. However, when designing our experiments, we considered the following: A major problem (or difficulty) with HPLC-MS/MS analyses is inter-run variability, which increases with the time between measurements (the more samples, the longer it takes). We therefore decided to keep the number of samples low enough to minimize inter-run variabilty – yet large enough to address our question. We decided for a compromise.

We do not understand „Were the CIA/Pg and Pg/CIA presented as one”? we analyzed two pg/CIA samples (open squares) and four pg/CIA samples (closed squares) and the cluster analysis was performed unsupervised.

--Data from the present study show important differences when compared with previous publications. For example, CIA groups exhibited almost no alveolar bone loss; few changes in joints were seen when Pg and Aa infection were performed before CIA. In contrast, previous studies account that arthritis per si induces alveolar bone loss and that Pg inoculation increases joint damage in CIA animals. Also, the lack of periodontal inflammation in Aa and Pg groups is dissimilar from previous literature. These differences should be discussed in details to substantiate further studies.

Agreed – and we revised the discussion accordingly. Please, see L 319-332

Reviewer #2: Buschhart et al examined the relationship between periodontal disease and inflammatory arthritis in two experimental protocols, in one the periodontal disease was induced prior to induction of arthritis, and in the other arthritis was induced prior to periodontal disease. The authors examined the outcome of the two diseases in these two experimental protocols.

The general take-home message is that, in these experimental set-ups, there was no effect of one disease on the other. These are still important findings to publish, however there are major concerns with the current version of the manuscript.

This reviewer found the manuscript very difficult to follow, and that is the gist of many of the comments below.

The other main problem is one of over-interpretation. Small differences that do not reach significance should not be presented as findings. This is true for the “slight” and “minor” changes found throughout – the major finding of the manuscript is that the diseases do not impact upon each other in mice under the experimental protocols presented.

Unfortunately, the difference in synovial proteome touted in the title and abstract, lead the reader to expect a more robust finding. In fact, the proteomic analysis was conducted on a small subset of the experimental groups, and the significance shown is a comparison to sham treated mice (no CIA, no pd) rather than to mice that did not receive the bacteria but were immunized to collagen. Therefore, here too the interpretation is not justified. This is especially true given the important negative finding that pd does not worsen arthritis.

We dare to contradict: when designing our experiments, we considered the following: A major problem (or difficulty) with HPLC-MS/MS analyses is inter-run variability, which increases with the time between measurements (the more samples, the longer it takes). We therefore decided to keep the number of samples low enough to minimize inter-run variabilty – yet large enough to address our question. We decided for a compromise – and after unsupervised clustering obtained quite clear and unexpected findings (explaining 70% of the variability!). Moreover: do our results contradict our own finding of pd not worsening arthritis? No – because in human RA, the severity of disease does by no means allow for defining the subtype of disease, leave alone a prognosis on which biological DMARD will lead to remission. Both, subtype of arthritis and response to biologicals are probably connected - yet utterly independent from disease severity. And in the best scenario, imagine you just need to identify the dominant oral pathobiont to decide on the most appropriate treatment option…..

And even though we do not consider us touting on any results, we deleted all “minor” and “slight” changes 

Specific comments:

The authors performed two independent experimental protocols, and show that nicely in figure 1. These protocols address independent questions! After that it is confusing whether we are being shown data from 1A or 1B or both – this has to be made very clear and it’s too confusing to show all joined together such as in 3B, especially since other figures like figure 4 are showing only the protocol of 1A. then figure 5 shows data for both protocols. It’s just too difficult to follow in this reviewer’s opinion. Best would be to show 1A and all the experimental evaluations of 1A, then show 1B and the evaluations of 1B. If there are comparisons between 1A and 1B (again these are independent questions, and performed independently), then that would come afterwards.

We fully agree that the first version of our manuscript was too dense – and we wrote the manuscript in this condensed form in order to save space. However, we are happy to follow your suggestion – and believe, that indeed, the present form is better and easier to follow.

Hard to follow the numbers in each group, and what was tested for each group. For example, figure 2A shows only sham vs. bacteria and the text provides the n of each group (large numbers for each group). Do these groups include +/-CIA? 

Again, we agree – and now provide separate graphs for all groups including the corresponding numbers of animals analyzed.

Figure 2B and 4C show heat maps but it’s unclear if the cytokine arrays were performed on all mice or only a select few mice per group. What was the actual group size for each endpoint measured? 

In fact, we only analyzed n=4 (randomly selected) for each group – so what you see is what we did. (the numbers are now provided in the corresponding Figure legends.

The heat maps also list individual mice not according to their group, further complicating the presentation. Some statistical comparison between groups is necessary to justify the conclusions that the authors present in the results section about induction or lack of induction of systemic inflammation.

We rephrased in Figure 5: “cluster 1 – mostly CIA” and “cluster 2 – mostly noCIA”. Beyond that, we are not exactly sure how to interpret your comment. What we did here is unsupervised clustering. A supervised ordering of samples within an hierarchically clustered heatmap is most unusual as it would interfere with the unsupervised data mining and even render it obsolete.

Line 135: It is unclear why the authors chose to analyze the gastrointestinal microbiome as a “proxy” for colonization of the oral cavity. They could have tested oral dysbiosis induced by Pg or Aa lavage – it has been shown by multiple groups that Pg induces an increase in the total anaerobic bacterial counts in the oral cavity detected by swabbing (no need to sacrifice the animals). Alternatively, they could have analyzed colonization of the oral cavity with the strains of bacteria they administered. Is there a reference for using the feces microbiome as a proxy for Pg or Aa colonization of the oral cavity? According to their results the gut microbiome did not reflect their intervention so what was learned? It’s confusing and misleading as presented. 

Well, we did not analyze the oral microbiome in live animals. Our mice would certainly not tolerate swabbing their mouth without additional anesthesia, which we did not get permission to do. We therefore wondered whether analyzing the gut microbiome would serve as a proxy (we rephrase in our manuscript) – which would present a non-invasive method to follow oral inoculation. In fact, Arimatsu and colleagues showed in 2014 that the ileum is the site to analyze however, that would require invasive procedures. What have we learned? That the gut microbiome/stool does not serve as a proxy. We are aware though, that not showing oral dysbiosis is a weakness and therefore will in future experiments analyze the oral microbiome and histology in the early phase of periodontal disease induction.

suggestion that bacteria induced a “minor” change in the gut microbiome, without showing statistical significance, is not justified. 

Agreed and rephrased

Furthermore, despite showing no dysbiosis induced by CIA, in the discussion line 299 the result is presented as if CIA indeed induced perturbation of the gut microbiome.

Agreed – even though we meant to say that antibiotics therapy perturbed the microbiome and thereby impacted on systemic cytokine production – we rephrased (L 313-316)

Line 185 reads “Figures 4A and B show macroscopic and histologic pictures of arthritic paws in mice that were subjected to periodontal disease induction before CIA” – actually what seems to be shown is one representative mouse of the control (no pd, no cia) group vs. one representative mouse of the CIA group (no pd). Therefore, the line is incorrect. 

Agreed and rephrased (please, see legend to Fig 4)

In fact, what needs to be shown is the scoring of the CIA in all groups (caliper measurements, incidence and severity).

We in fact gave up on caliper measurements some time ago, as they are less informative than our visual scoring (because the caliper does not allow for reproducible measurements of individual digits). Moreover, calipers require fixation of the animals – which we circumvent with the visual scoring. In summary, a caliper would only confirm the visual scoring of the metacarpal joints – but would not provide any additional information.

Figure 3b should compare independently the CIA+bacteria group to the CIA-bacteria, and the CMC-CIA to bacteria-CIA. Presently the comparison groups together the +/-CIA groups in the sham vs Pg or Aas with and without CIA.

Agreed and done

Minor:

The legends appear in the text rather than at the end.

This is what the PLOS ONE formatting guidelines ask for

Line 395-400: clarify the method – are you administering the bacteria by gavage (in which case 25 ul does not seem realistic, and no gavage needle size is mentioned), or did you “apply” the bacteria? If applied, clarify what you mean.

We “applied” the bacteria, rephrased throughout the manuscript and specified the materials and methods section.

Line 401 – what is “initial” about this?

Agreed – and removed

Line 112 – as far as I understand it has not been demonstrated that systemic abx are necessary and indeed “facilitate colonization.” If you have a reference for this, please include. If not, please omit the claim. My understanding is that the model employs systemic abx based on the hypothesis that abx will facilitate colonization but that when it was tested the abx were shown not to be necessary.

Agreed – we also did not find proof that you need systemic abx to facilitate colonization – and therefore omit the “claim” (see first paragraph of the results section L 77/78)

Line 186 – before jumping to cytokines the authors should say something about the arthritis incidence and scores – these should be shown.

Agreed and done

Last line of introduction – change synoviale to synovial

Done

Line 412 “tree” should be “three”

done

---

## [Editor Report · Decision Letter 1]

11 Nov 2020

Periodontal pathogens alter the synovial proteome

Periodontal pathogens do not exacerbate macroscopic arthritis but alter the synovial proteome in mice

PONE-D-20-26647R1

Dear Dr. Müller-Hilke,

We’re pleased to inform you that your manuscript has been judged scientifically suitable for publication and will be formally accepted for publication once it meets all outstanding technical requirements.

Kind regards,

David Douglass Brand

Academic Editor

PLOS ONE

Additional Editor Comments (optional):

The authors have made a very significant effort toward responding to the criticisms provided by both reviewers as well as those I myself made in the primary review. I would agree with Dr. Müller-Hilke that the revisions have strengthened the work considerably. One final comment on this work is that the use of Tramadol, while an admirable attempt to relieve the pain experienced by the mice in the study, may have a significant effect on the study overall in that it is known to inhibit experimental inflammation (Eur J Pain. 2000;4(4):413-5. doi: 10.1053/eujp.2000.0208) the very endpoint that is measured in collagen-induced arthritis. This should be pointed out to the committee(s) that oversee animal welfare considerations and should be taken into consideration in the design of any future experiments that involve an inflammatory response.
---

## [Editor Report · Acceptance letter]

16 Dec 2020

PONE-D-20-26647R1 

Periodontal pathogens alter the synovial proteomePeriodontal pathogens do not exacerbate macroscopic arthritis but alter the synovial proteome in mice 

Dear Dr. Müller-Hilke:

I'm pleased to inform you that your manuscript has been deemed suitable for publication in PLOS ONE. Congratulations! Your manuscript is now with our production department. 

Kind regards, 

on behalf of

Dr. David Douglass Brand 

Academic Editor

PLOS ONE